# Protective Factors, Risk of Violence and Discrimination and Mental Health Indicators of Young LGB People

**DOI:** 10.3390/ijerph192114401

**Published:** 2022-11-03

**Authors:** Marta Evelia Aparicio-García, Eva M. Díaz-Ramiro, Susana Rubio-Valdehita, M. Inmaculada López-Núñez, Isidro García-Nieto

**Affiliations:** 1Instituto de Estudios Feministas, Universidad Complutense de Madrid, 28040 Madrid, Spain; 2Social, Work and Differential Department, Facultad de Psicología, Universidad Complutense de Madrid, 28223 Madrid, Spain; 3Servicio LGTBI de la Comunidad de Madrid, 28040 Madrid, Spain

**Keywords:** adolescent, health, sexual orientation

## Abstract

Lesbian, gay and bisexual (LGB) people have more risk of suffering from violence and situations of discrimination than heterosexual people. The current study compares LGB people with heterosexual people in protective factors, violence, health and well-being factors. The sample comprises 609 Spanish people between 14 and 25 years old. We established a cross-sectional design. A survey including questions about sociodemographic information and protective, violence and health and well-being factors was designed ad hoc for this study. The results show that the LGB group (*n* = 342) is more at risk of verbal and physical violence and feels more isolated than the heterosexual participants (*n* = 267). In contrast, heterosexual participants report having more employment discrimination. No significant differences were found in social support or psychological health. These results are important to understand the state of social normalization and non-discrimination for LGB people in certain contexts in Spain, and its impact on psychological health.

## 1. Introduction

The American Psychological Association refers to sexual orientation as an enduring pattern of emotional, romantic and/or sexual attractions to men, women or both sexes. Sexual orientation also refers to a person’s sense of identity based on those attractions, related behaviors, and membership in a community of others who share those attractions [1]. Research over several decades has demonstrated that sexual orientation ranges along a continuum, from exclusive attraction to the other sex to exclusive attraction to the same sex. However, sexual orientation is usually discussed in terms of three categories: heterosexual (attractions to members of the other sex), gay/lesbian (attractions to members of one’s own sex) and bisexual (attractions to both men and women) [2].

The health of sexual minority people has been stigmatized for many years, until the emergence of Meyer’s model [3], in which the perspective on their health changed. The Minority Stress Model postulates that sexual minorities experience increased mental health problems because of stress processes unique to their status, namely discrimination, expectations of rejection, concealment, and internalized homophobia. Integrating this model and the Psychological Mediation Framework [4], the Health Equity Promotion Model [5] adds a life course development perspective within a health equity framework to highlight how social positions (for example, socioeconomic status, age, race/ethnicity, place of residence, religiosity, etc.), individual, structural and environmental contexts (social exclusion, discrimination, and victimization) and health-promoting and adverse pathways (behavioral, social, psychological, and biological processes) influence the continuum of health outcomes in LGB communities.

The first level of these models refers to social position. LGB communities are socially heterogeneous, so intersectional variables related to social position must be considered, for example, gender, age, socioeconomic status, race/ethnicity, place of residence, religiosity, etc. [3]. The second level, individual, structural and environmental context, shows that LGB youth are exposed to discrimination, victimization and social exclusion [6,7]. The third level refers to health promotion and adverse pathways identified in LGB people as a group at psychological risk: higher rates of depression [8], anxiety [9], suicide ideas and attempts [10] and increased drug and alcohol use compared to heterosexual youth [11].

However, there are important individual differences in the health of young LGB individuals, indicating the existence of specific protective factors that affect health despite the risk [12], such as the Health Equity Promotion Model proposes [5]. The protective mechanisms of young LGB people against homophobia are grouped into personal, relational and contextual resources [13]. Identification with the LGB community acts as a predictor of lower anxiety, depression and perceived stress [14] and with higher self-esteem [15]. Research indicates that the perception of a low acceptance is associated with the appearance of suicidal ideas among the young LGB individuals [16]. Having at least four friends of the same sexual orientation acts as a modulator that reduces the risk of alcohol consumption in young people who drink to escape from perceived discrimination [17]. Social and family support is a protective mechanism related to low levels in depressive symptoms [18] reaching the same levels of depression between LGB and heterosexual youth [19]. Other authors point out that social or family support is a predictor of self-esteem [20]. Family relationships with low support and little union are associated with increased drug use and running away from home. Specifically, in lesbian and bisexual girls, running away from home showed positive correlation with the perception of low parental support, while in gay and bisexual boys, in addition to the low perceived support, it was observed that participation in activities with the family was associated with increased drug consumption and running away from home, while this engagement had a protective effect for heterosexual boys [21]. Research indicates that the perception of low acceptance is associated with suicidal ideas among the LGB youth [16].

The situation of lesbians, gay and bisexual people (LGB) in Spain has improved very much in the last few decades. A report shows that 88% of Spaniards think that society should accept the relationship between same-sex people (this is the highest rate of acceptance among the 39 analyzed countries) [22]. Even with the existence of some legislation that acknowledges LGB rights and advocates for more social tolerance, there is also abundant evidence that Spanish society still exhibits prejudice and discrimination. In 2013, the State Federation of Lesbians, Gays, Transsexuals and Bisexuals (FELGTB) and the Lesbian, Gay, Transgender and Bisexual Association of Madrid (COGAM) conducted a state-level study by collecting 762 valid population questionnaires among LGBT. From this sample, 45% claimed to have been discriminated against because of their sexual orientation or gender identity in situations of their daily lives (in a restaurant, bar or similar, in establishments open to the public, at the time of renting a house, etc.…), 31.23% in jobs (most of them by jokes or discriminatory treatment by their colleagues) and sexuality [23]. In 2017, 629 incidents of hate due to sexual orientation or gender identity were reported in Spain [24]. A study points out the importance of analyzing data from different countries to analyze the risk of violence and safety for LGB people, since the data are still contradictory and scarce [7]. However, the data in Spain are focused on the adult population and not on young people, who are at high risk of suffering victimization in socialization environments [6].

Based on previous models, the purpose of the current study is to analyze the influence on LGB youth and heterosexual youth in the three dimensions: social position; individual, structural and environmental context; and health and adverse pathways. Four hypotheses are evaluated: (1) LGB youth have more risk of violence and discrimination; (2) LGB youth have more risk of trying drugs, alcohol and tobacco, and practice less sports than heterosexual individuals; (3) LGB youth receive less support from family and friends and participate less in social activities than heterosexual youth; (4) LGB people have worse mental health indicators than heterosexual people.

## 2. Materials and Methods

### 2.1. Participants

Participation in the study was offered to 795 young people between 14 and 25 years old, and 745 responded to the survey (93.71% response rate). The youth population comprises individuals between 15 and 24 years of age according to resolution 36/28 adopted by the United Nations General Assembly in 1981. Of that population, only lesbians, gays, heterosexuals and bisexuals were included in the analysis, leaving a total sample of 609 participants for this study (81.44% of all respondents). The average age of participants was 20.59 years (*SD* = 3.14). A total of 211 were identified as lesbian/gay (34.7%), 131 as bisexual (21.5%) and 267 as heterosexual (43.8%). As for the religion of the participants, 77.5% were atheist or agnostic, 17.0% Catholic and 5.5% followed another religion. Most of the participants were living in an urban zone (87.5%), and only 12.5% resided in a rural zone (Table 1).

### 2.2. Instrument

To assess the different levels of the cited theorical models, an ad hoc survey was built for this research. The survey was divided in three areas. The first part of the survey included questions about *individual variables*, such as gender, age, sexual orientation (“Identify your sexual orientation as”, with response options of “heterosexual”, “gay”, “lesbian” or “bisexual”), residence area (“Would you describe the area in which you live?”, with response options of “rural” or “urban”), and religion (“What is your religion?”, with the response options “Catholic”, “Jewish”, “Muslim”, “Evangelist”, “Other”, “Atheist”, “Agnostic” or “I prefer not to answer”).

The second level was *environmental contexts* with 6 questions about social exclusion, discrimination and victimization (e.g., “*I have been verbally harassed at school*”). 

The third area dealt with health and had 16 questions divided by: behavioral level with 6 questions (e.g., “I practice a sport in school or outside it”); social and community level with 6 questions (e.g., “I have support from my family”), and psychological processes with 4 questions (e.g., “Once I have thought about suicide”). To obtain a mental health indicator, the General Health Questionnaire (GHQ-12) [25] was included at the end of the survey. The GHQ-12 has demonstrated adequate reliability and validity to detect mental health problems and to assess psychological distress in the Spanish population [26].

### 2.3. Procedure

Participants were recruited via a snowball sampling procedure by posting the research description, the survey link on Google Docs (Android version 1.6.292), the contact of the principal researchers on Twitter, Facebook and other social networking websites and different associations (Daniela Foundation, FELGTB and COGAM) and mailing the description to high school institutions in Spain. We contacted these organizations and explained the goals of the research and the method and explained the purpose of our research.

### 2.4. Ethical Procedures

The protocol for the present study obtained approval from the Ethics Committee of Complutense University of Madrid (Ref: CE_20211216-06_SOC). All participants read a brief instruction describing the research and agreed to participate before answering the survey. Participants in the research were anonymous and voluntary, and we asked about “consent to participate”.

### 2.5. Data Analysis

Associations between sexual orientation and multi-level context, health-promoting and adverse pathways and mental health indicators were analyzed by logistic regression using SPSS 22.0 version (IBM, Madrid, Spain). Participant’s age, residence and religion were included as covariates in the logistic regression models to control a possible moderating effect. Prior to logistic regression, associations between residence and religion and sexual orientation were evaluated by Pearson’s χ^2^, and in addition, average ages between groups were compared by computing *F* statistic. The global punctuation of GHQ-12 was dichotomized in whether or not they have psychological problems, taking into account the cut-off points of the Spanish population [26]. Heterosexual was included in all analysis as the reference group.

## 3. Results

The variables included to assess social position show a significant association between religion and sexual orientation (*χ^2^* = 11.89, *p* = 0.018), but the relationship between residence zone and sexual orientation was not significant (*χ^2^* = 1.49, *p* = 0.476). Significant differences in age between groups were found [*F* (2608) = 5.08, *p* = 0.007). As can be seen in Table 1, the gay and lesbian group was the oldest.

Regarding the assessment of the risk of discrimination and violence of LGBT youth (Hypothesis 1), the gay and lesbian group showed more risk of verbal and physical violence out of school (Table 2).

In behavioral risk analysis (Hypothesis 2), heterosexual participants showed greater participation in sports in school or outside (Table 3). No differences were found between the groups in drug use, tobacco use or alcohol use.

In the social support variables (Hypothesis 3), only taking part in LGBT associations were significant (Table 4). In most cases, heterosexuals showed greater participation in the different activities considered. No significant differences were found in relation to the social support received from family, adults or friends.

In terms of mental health indicators (Hypothesis 4), LGB groups showed a significantly higher percentage of people who feel isolated compared to the heterosexual group (Table 5), but the rest of the variables show the same levels as for heterosexuals.

## 4. Discussion

In our study, we analyzed perceived violence and discrimination by assessing risk behaviors, social support and mental health in three groups based on sexual orientation (heterosexual, gay/lesbian and bisexual) of young people in Spain.

Regarding the factors of violence and discrimination (first hypothesis), LGB people showed a higher risk of violence in terms of more verbal and physical attacks outside of school. Previous studies have pointed out that LGB people experience a relatively high degree of violence and abuse and often are victims of hate crimes [27]. LGB people may experience various forms of discrimination, including ageism, heterosexism and homophobia which will lead to marginalization and invisibility [28,29,30]. However, there are not differences in other variables. With the recognition of equal marriage in Spain in 2005, as well as information campaigns about the reality of the LGB collective, perceptions about lesbians, gays and bisexuals have improved Therefore, discrimination and violence have decreased. Previous studies assessed discrimination from a wider perspective (social or familiar) [31]. This survey shows that discrimination in Spain is somewhat below the average of all European countries, with 16% of the LGB population claiming to have felt discriminated against. Spain ranks seventh in discrimination among 28 European countries. This result contrasts considerably with that offered by the survey conducted by the FELGTB and the COGAM [23], which showed a 31.23% rate of discrimination, almost twice as much as in the FRA survey. In the 2015 Eurobarometer on discrimination, aimed at the general population—not only LGBT people—when asked about their personal attitude, more specifically how they would feel comfortable with having LGBT people as their peers, the majority responded that they would feel totally or moderately comfortable, 81% with LGB partners and 78% if they were transexual peers. The majority percentages are consistent with the idea that discrimination is due to a minority of people having “homophobic profiles”. Indeed, 4–6% of individuals recognize that they would feel uncomfortable, a percentage that although reduced, still maintains the potential to generate situations of discrimination against LGB people.

In terms of risk behaviors (second hypothesis), LGB people practice less sports than heterosexual people, but there are not differences in the rest of the indicators (tobacco, alcohol and drug use). Other authors have pointed out that LGB people reported less participation in physical activities [32]. However, previous studies show that LGB people use more alcohol, tobacco and drugs [33,34]. In an Australian study with the general population, there were elevated rates of past year cannabis (22.4%), ecstasy (11.8%) and methamphetamine (9.7%) use among GB men compared to heterosexual men (12.4%, 2.9% and 2.5%). LB women also reported elevated rates of past-year use (tobacco—23.7%; cannabis—24.6%) compared to heterosexual women (10.6% and 7.1%). LB women initiated tobacco (15.2 years) and alcohol (15.5 years) at an earlier age than heterosexual women (16.6 and 17.7 years) and were significantly more likely to report daily alcohol consumption (OR 3.2, 95% CI: 2.1, 5.1), and weekly or more frequent cannabis use (OR 1.7, 95% CI: 1.1, 3.1) [35].

Regarding social support (third hypothesis), LGB people have less support from their families and adults than heterosexual people; however, the differences were not statistically significant. Other studies show that lesbian and bisexual women report lower levels of parental support than heterosexual women and that gay men report lower levels of parental support than bisexual and heterosexual men [36] and family support predicts better general health status [37].

In terms of health and well-being indicators (fourth hypothesis), LGB have more risk of feeling isolated than heterosexual people, but there are not differences in the rest of the indicators (suicidal ideas, psychological health problems, and feeling of happiness). Previous studies showed that LGB people are more at risk of suicide and more mental health problems [33,34,38]. However, in our study we obtained good results because LGB people did not show risk behavior in these indicators, although they feel isolated. It is necessary to evaluate this aspect more because there are not differences in support perceived for LGB people (Table 4), but they feel isolated (Table 5).

Our study has some limitations that should be overcome in future studies. First, despite the number of participants, the sample should be expanded to improve the generalizability of the results. We must not forget that the information was obtained through the collaboration of an LGB association (although not alone), and this could bias the results, leaving people that have no Internet access or who are not involved in the associative sector out of the study [39]. In addition, people of other sexual orientations, such as pansexual, demi sexual, asexual, etc., have been excluded from the inclusion criteria of the sample. It would be very interesting to consider this group of people in future research and analyze their health outcomes. Secondly, in our study we asked about the variables of the study through a survey, but not with a complete questionnaire that measured each of the variables, so it would be advisable in future research to use a broader questionnaire to deepen the answers of participants, because there could be a certain acquiescence in yes/no responses. Thirdly, as the survey contained many sensitive topics, young people could choose not to answer a question. This could be the reason why there was a small percentage of missing data. Finally, due to the fact our study is quantitative, some questions have not been resolved; for example, why LGB people practice less sports or feel more isolated. It would be interesting to analyze in future studies the reasons why they do not participate as much as the other groups through a qualitative study.

## 5. Conclusions

Multiples studies have analyzed the health and the discrimination in LGB people; however, in Spain there are no studies that analyzed the perceived violence, discrimination and well-being of young LGB people compared with heterosexual youth. Our results are consistent with previous studies when pointing out that LGB youths have more risk of feeling isolated than heterosexuals and show a higher risk of violence. The results indicate that the health of Spanish LGB young people is not particularly worse than that of heterosexual young people in terms of social support, suicide and emotional well-being, probably explained by the policies established in recent years in Spain. The importance of dealing with sexuality and all the psychological processes involved in accepting one’s own identity by recognizing oneself as lesbian, gay, bisexual are intimately deep processes where the person is highly vulnerable. Accepting sexual orientation and accepting one’s own stigma and overcoming it, are processes that deserve to be reinforced, especially among the population working with young people. There is a need for comprehensive and inclusive sex education in educational institutions to be a point of reference in high schools and primary schools where bullying of this population is prevented, as well as more training in LGTBIQ issues, sexuality and mental health, with a holistic and biopsychosocial perspective.

## Figures and Tables

**Table 1 ijerph-19-14401-t001:** Demographic characteristics of participants by sexual orientation.

	Heterosexual	Gay/Lesbian	Bisexual
	*n* (%)	*n* (%)	*n* (%)
Residence			
Rural	38 (14.2)	23 (10.9)	15 (11.5)
Urban	226 (84.6)	188 (89.1)	116 (88.5)
Missing	3 (1.1)	0 (0.0)	0 (0.0)
Religion			
Catholic	57 (21.4)	28 (13.3)	14 (10.7)
Atheist/agnostic	180 (67.4)	162 (76.8)	109 (83.2)
Other	14 (5.2)	13 (6.2)	5 (3.8)
Missing	16 (6.0)	8 (3.7)	3 (2.3)
Age			
Mean (SD)	20.25 (3.24)	21.14 (3.08)	20.40 (2.93)

**Table 2 ijerph-19-14401-t002:** Associations between sexual orientation and individual and structural level.

	*n* (%)	Odds Ratio (95% Confidence Interval)	*p*
Structural level			0.632
Excluded by your peer group at some time			
Heterosexual (*n* = 245)	52 (21.2%)	1.0	
Gay/Lesbian (*n* = 201)	48 (23.9%)	1.25 (0.78–1.99)	
Bisexual (*n*= 118)	27 (22.9%)	1.17 (0.68–2.01)	
Individual level			
Verbal attacks at school			0.871
Heterosexual (*n* = 262)	74 (28.2%)	1.0	
Gay/Lesbian (*n* = 207)	55 (26.6%)	1.04 (0.67–1.59)	
Bisexual (*n*= 127)	33 (26.0%)	0.90 (0.55–1.49)	
Verbal attacks out of school			0.000
Heterosexual (*n* = 265)	66 (24.9%)	1.0	
Gay/Lesbian (*n* = 207)	88 (42.5%)	2.38 (1.58–3.59)	
Bisexual (*n*= 129)	40 (31.0%)	1.36 (0.84–2.21)	
Physical attacks at school			0.809
Heterosexual (*n* = 265)	33 (12.5%)	1.0	
Gay/Lesbian (*n* = 207)	18 (8.7%)	0.82 (0.44–1.54)	
Bisexual (*n*= 127)	13 (10.2%)	0.86 (0.42–1.78)	
Physical attacks out of school			0.008
Heterosexual (*n* = 94)	12 (12.8%)	1.0	
Gay/Lesbian (*n* =205)	53 (25.9%)	3.02 (1.39–6.56)	
Bisexual (*n*= 122)	23 (18.9%)	1.98 (0.83–4.46)	
Cyberbullying			0.059
Heterosexual (*n* = 242)	51 (21.1%)	1.0	
Gay/Lesbian (*n* = 198)	57 (28.8%)	1.73 (1.10–2.74)	
Bisexual (*n*= 124)	32 (25.8%)	1.31 (0.77–2.22)	

**Table 3 ijerph-19-14401-t003:** Associations between sexual orientation and health factors.

	*n* (%)	Odds Ratio (95% Confidence Interval)	*p*
Practice of a sport in school or outside it			0.002
Heterosexual (*n* =253)	143 (56.6%)	1.0	
Gay/Lesbian (*n* = 200)	100 (50.0%)	0.79 (0.53–1.18)	
Bisexual (*n*= 121)	42 (34.7%)	0.44 (0.28–0.70)	
Tried drugs and alcohol			0.145
Heterosexual (*n* = 261)	130 (49.8%)	1.0	
Gay/Lesbian (*n* = 208)	125 (60.1%)	1.39 (0.94–2.05)	
Bisexual (*n*= 129)	79 (61.2%)	1.44 (0.92–2.26)	
Smoke			0.860
Heterosexual (*n* = 267)	59 (22.1%)	1.0	
Gay/Lesbian (*n* = 210)	53 (25.2%)	1.11 (0.72–1.73)	
Bisexual (*n*= 130)	33 (25.4%)	1.12 (0.68–1.85)	
Drank alcohol			0.256
Heterosexual (*n* = 266)	151 (56.8%)	1.0	
Gay/Lesbian (*n* = 210)	141 (67.1%)	1.39 (0.93–2.08)	
Bisexual (*n*= 131)	78 (59.5%)	1.06 (0.68–1.67)	
Drug use			0.258
Heterosexual (*n* = 267)	59 (22.1%)	1.0	
Gay/Lesbian (*n* = 135)	59 (28.1%)	1.21 (0.79–1.86)	
Bisexual (*n*= 131)	41 (31.3%)	1.50 (0.92–2.42)	

**Table 4 ijerph-19-14401-t004:** Associations between sexual orientation and social support.

	*n* (%)	Odds Ratio (95% Confidence Interval)	*p*
Friends support			0.653
Heterosexual (*n* = 82)	74 (90.2%)	1.0	
Gay/Lesbian (*n* = 200)	189 (94.5%)	1.57 (0.58–4.23)	
Bisexual (*n*= 120)	111 (92.5%)	1.18 (0.42–3.28)	
Bisexual (*n*= 123)	28 (22.8%)	1.39 (0.78–2.48)	
Go out with friends			0.316
Heterosexual (*n* =267)	224 (83.9%)	1.0	
Gay/Lesbian (*n* =209)	178 (85.2%)	0.90 (0.53–1.53)	
Bisexual (*n*= 131)	104 (79.4%)	0.64 (0.36–1.14)	
Family support			0.437
Heterosexual (*n* = 82)	47 (57.3%)	1.0	
Gay/Lesbian (*n* = 187)	101 (54.0%)	0.74 (0.42–1.32)	
Bisexual (*n*= 122)	62 (50.8%)	0.68 (0.37–1.24)	
Adult support outside the family			0.349
Heterosexual (*n* = 82)	51 (62.2%)	1.0	
Gay/Lesbian (*n* = 180)	109 (60.6%)	0.73 (0.40–1.32)	
Bisexual (*n*= 111)	61 (55.0%)	0.63 (0.34–1.18)	
Involvement in extracurricular activities in school			0.954
Heterosexual (*n* = 209)	92 (44.0%)	1.0	
Gay/Lesbian (*n* = 167)	70 (41.9%)	1.03 (0.67–1.59)	
Bisexual (*n*= 101)	40 (39.6%)	0.95 (0.58–1.57)	
Take part in LGBT associations			0.000
Heterosexual (*n* =73)	38 (52.1%)	1.0	
Gay/Lesbian (*n* = 178)	53 (29.8%)	0.38 (0.20–0.71)	
Bisexual (*n*= 112)	23 (20.5%)	0.23 (0.11–0.40)	
Take part in online LGTB groups			0.345
Heterosexual (*n* = 79)	35 (44.3%)	1.0	
Gay/Lesbian (*n* = 206)	65 (31.6%)	0.65 (0.36–1.14)	
Bisexual (*n*= 126)	42 (33.3%)	0.70 (0.37–1.29)	

**Table 5 ijerph-19-14401-t005:** Associations between sexual orientation and mental health indicators.

	*n* (%)	Odds Ratio (95% Confidence Interval)	*p*
Feeling isolated			0.000
Heterosexual (*n* = 261)	52 (19.9%)	1.0	
Gay/Lesbian (*n* = 106)	42 (39.6%)	2.80 (1.66–4.71)	
Bisexual (*n*= 83)	41 (49.4%)	4.03 (2.32–7.00)	
Ever think about suicide			0.334
Heterosexual (*n* = 257)	115 (44.7%)	1.0	
Gay/Lesbian (*n* = 206)	87 (42.2%)	1.00 (0.68–1.48)	
Bisexual (*n*= 130)	70 (53.8%)	1.36 (0.87–2.12)	
Happy or very happy			0.724
Heterosexual (*n* =247)	203 (82.2%)	1.0	
Gay/Lesbian (*n* =201)	166 (82.6%)	0.94 (0.57–1.58)	
Bisexual (*n* = 117)	92 (78.6%)	0.79 (0.45–1.40)	
Psychological health problems (GHQ-12)			0.911
Heterosexual (*n* = 267)	109 (40.8%)	1.0	
Gay/Lesbian (*n* = 211)	83 (39.3%)	0.92 (0.63–1.35)	
Bisexual (*n*= 129)	53 (41.1%)	0.98 (0.63–1.52)	

## Data Availability

The data presented in this study are available on request from the corresponding author.

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
