# Peer review of "Protective Factors, Risk of Violence and Discrimination and Mental Health Indicators of Young LGB People"

_ijerph, 2022, doi:10.3390/ijerph192114401_

Round 1

Author Response

The comments are attached

Reviewer 2 Report

The article focuses on an important topic, taking into account a specific country, Spain.

There are some critical points that the authors need to take into account to improve the paper. I believe that the proposal is valuable although it needs some relevant revision in order to be published: 

- First of all, the research focuses on "young" people. who falls into the youth category? is this a generational study? or was the age group used considering some official statistical source? The authors should clarify this choice. 

- The paper has not a proper theoretical part that could be useful to create a theoretical framework useful to read (and to position) the paper.

- The intro is not informative. There are lots of "unnecessary" information, such as historical anecdotes that could be interesting to introduce the topic but should be trimmed and supported by the theoretical framework.

- The methodological part is not informative. The methodological choices are not well supported. for example, why and how the authors used digital methods? We cannot forget that online survey is different from ftf research. It should be important to problematize it. On this topic I suggest using for example:

Monaco, S. (2022). Gender and sexual minority research in the digital society. In Handbook of research on advanced research methodologies for a digital society (pp. 885-897). IGI Global.

- In the Discussion part there are lots of interesting ideas, but there is not a strong connection between the different parts.

- Also conclusions are not meaningful because there is a lack of a strong idea underpinning the article.

Author Response

The comments are attached
